# *In Vitro*, *Ex Vivo* and *In Vivo* Techniques to Study Neuronal Migration in the Developing Cerebral Cortex

**DOI:** 10.3390/brainsci7050048

**Published:** 2017-04-27

**Authors:** Roberta Azzarelli, Roberto Oleari, Antonella Lettieri, Valentina Andre', Anna Cariboni

**Affiliations:** 1Department of Oncology, University of Cambridge, Hutchison-MRC Research Centre, Hills Road, Cambridge CB2 0XZ, UK; RA461@cam.ac.uk; 2Wellcome Trust—Medical Research Council Cambridge Stem Cell Institute, University of Cambridge, Tennis Court Road, Cambridge CB2 1QR, UK; 3Cavendish Laboratory, Department of Physics, University of Cambridge, Cambridge CB3 0HE, UK; 4Department of Pharmacological and Biomolecular Sciences, University of Milan, Via Balzaretti, 9, Milan 20133, Italy; roberto.oleari@unimi.it (R.O.); antonella.lettieri@unimi.it (A.L.); valentina.andre@unimi.it (V.A.); 5Institute of Ophthalmology, University College London, 11-43 Bath Street, London EC1V 9EL, UK

**Keywords:** migrating neuron, cortex, migration assays, electroporation

## Abstract

Neuronal migration is a fundamental biological process that underlies proper brain development and neuronal circuit formation. In the developing cerebral cortex, distinct neuronal populations, producing excitatory, inhibitory and modulatory neurotransmitters, are generated in different germinative areas and migrate along various routes to reach their final positions within the cortex. Different technical approaches and experimental models have been adopted to study the mechanisms regulating neuronal migration in the cortex. In this review, we will discuss the most common in vitro, ex vivo and in vivo techniques to visualize and study cortical neuronal migration.

## 1. Introduction

Cell migration is essential for the development and maintenance of multicellular organisms and it is also involved in pathological events such as inflammation and cancer [1,2]. In the brain, newborn neurons have to migrate from their site of origin into their final positions to integrate into functional neuronal circuits.

Neuronal migration is a complex process that involves a plethora of intrinsic and extrinsic factors that strictly cooperate to control the different phases and aspects of cell movement. These include transcription factors, adhesion molecules, secreted cues and extracellular matrix components. Within the cell, extracellular stimuli converge in the regulation of the cytoskeleton and ultimately in control of cell movement. Different modes of migration have been described in the brain, including radial, tangential and axonophilic migration [3]. The cerebral cortex represents an excellent model to investigate neuronal migration, because both radial and tangential modes of migration contribute to cortex formation [3]. Thus, in this review we will focus on the developing cerebral cortex and illustrate the experimental approaches that can be adopted to study the molecular mechanisms that control cortical neuronal migration in mouse models.

## 2. Neuronal Migration in the Developing Cerebral Cortex

The cerebral cortex is the site of higher human cognitive functions such as memory, consciousness, fine movements and thoughts and it exhibits a stereotypic histological organization, with neurons arranged into six horizontal layers [4,5,6]. This unique cytoarchitecture is established during embryonic development through the coordinated processes of neurogenesis and cell migration. The mechanisms that control the development of cerebral cortex have been extensively studied in rodents and in particular in the mouse. Indeed, the basic mechanisms of cortical development and the overall cortical arealization and cellular distribution are conserved between mice and humans and the relatively fast brain development in mice makes them very useful models. However, it is important to note that major differences, such as in size and gyrification, distinguish the human from the rodent brain and thus some principles of neurogenesis might differ [7].

During mammalian embryonic development, newborn neurons leave the proliferative areas to reach the place of final maturation in precise locations within the cortex. Two main neuronal subtypes populate the cerebral cortex: excitatory pyramidal neurons and inhibitory interneurons [6,8]. Excitatory neurons represent the majority of all cortical neurons and they use glutamate as their neurotransmitter. These neurons are born within the dorsal brain in the areas close to the internal lateral ventricles, called the ventricular (VZ) and subventricular zones (SVZ). Soon after birth, glutamatergic neurons migrate from the proliferative areas in an upward direction towards the cortical plate, using radial glia-guided migration (Figure 1) [9]. Instead, inhibitory interneurons comprise only a small fraction (Ca 20%) of all cortical neurons and release the neurotransmitter gamma-aminobutyric acid (GABA). Inhibitory GABAergic neurons are born in the ventral regions of the brain and they have to migrate long distances to reach the cortex using a mode of migration called tangential migration (Figure 1) [3,10]. Although the distinction in radially migrating glutamatergic neurons and tangentially migrating interneurons is particularly helpful for discussion purposes, it is important to note that several classes of glutamatergic neurons, such as Cajal Retzius cells, a subpopulation of subplate neurons and transient cortical neurons also migrate tangentially to disperse along the medio-lateral axis [11].

Disruption of neuronal migration can be easily studied in mice and several research studies have shed light on the genes and the molecular mechanisms underlying cortical malformations in humans. The consequences of such migration abnormalities include severe mental retardation, epilepsy and various intellectual disabilities [12,13].

When radial neuronal migration is perturbed during embryonic development, neurons do not reach their correct final location in the cortical plate (CP), but they remain either close to the ventricle (periventricular heterotopia PH) or in the white matter beneath the CP (subcortical band heterotopia SBH) [14]. In some cases, neurons are still able to migrate to the cortical plate. However, they generate a very thick and disorganized cortex. As a consequence, the normal pattern of gyrification of the human brain is disrupted, leading to a simplified (pachygyria) or absent (agyria) degree of convolutions [13]. Agyria and pachygyria are typical features of a common neuronal migration disorder, called lissencephaly (LIS), which owes its name to the smooth appearance of the brain surface.

Defects in tangential migration of cortical interneurons may result in inappropriate and unbalanced neuronal circuit formation [3,15]. Such wiring defects are thought to be involved in the etiology of various neurological disorders, including autism, schizophrenia and epilepsy and it is, therefore, important to further investigate the molecular mechanisms regulating tangential migration. Below we will discuss the biology of the two main different modes of migration in the cerebral cortex: radial and tangential migration.

### 2.1. Radial Migration

The term radial migration came from the discovery that newborn neurons migrate to the cortical plate using the radial glia filaments as a scaffold. Radial glia cells (RGCs) are neural progenitor/stem cells that reside in the VZ at mid-late stages of corticogenesis [16]. RGCs divide at the apical surface (VZ), are capable of self-renewal and generate most of the cortical neurons and the neuronal committed intermediate progenitors [17]. More importantly, RGCs exhibit a typical bipolar shape, with their nucleus predominantly found in the VZ, a short apical process that connects them with the lateral ventricles and a very long basal filament that contacts the outer brain surface. This array of basal radial glia filaments is what newborn neurons use to support and direct their migration towards the outer layers of the cortical plate. As neurons start to settle, the cortical wall thickness increases and later born neurons will migrate past the earlier born neurons and progressively populate more superficial layers in an “inside-out” fashion. This highly coordinated process generates the stereotypic six-layered organization of the cortex, whereby neurons in each layer are born at the same time and exhibit similar molecular, morphological and hodological (connection pattern) properties [5].

### 2.2. Tangential Migration

Tangential migration is the predominant mode of migration of GABAergic cortical interneurons, which originate from the ventral brain, in the VZ and SVZ of the medial (MGE), lateral (LGE) and caudal (CGE) ganglionic eminences [3,10,18]. In addition, recent work has shown that roughly 10% of cortical interneurons originate from the Preoptic area (POA) and migrate long distances to reach the cortex [19].

Cortical interneurons travel via distinct tangential routes in order to enter the cortex: at early developmental stages, young neurons born in the MGE circumvent the striatum to enter the cortex via two routes, the superficial marginal zone and the deeper IZ-SVZ [3]. Later in development, the LGE and the dorso-caudal GE (dCGE) also contribute to the generation of cortical interneurons and these interneurons enter the cortex only via the SVZ and IZ [10].

Once they have reached the cortex, interneurons integrate into the cortical plate to make local connections with excitatory pyramidal cells. To reach their final laminar position in the cortex, interneurons switch their mode of migration from tangential to radial, although they do not strictly rely on radial glia fibers [20,21]. Indeed, the leading process of radially migrating interneurons is not always aligned with the glial process, but it also makes contacts with axons or dendrites of pyramidal neurons [22]. Interestingly, recent work has suggested that, similar to what described for pyramidal neurons, cortical interneurons also distribute across the cortical layers according to their date of birth [23].

## 3. How to Study Neuronal Migration in the Cerebral Cortex

Research on the molecular mechanisms regulating cortical neuronal migration is relevant not only to understand brain development, but also to gain insights into the mechanisms that underlie several neurological disorders. The experimental approaches that we will discuss in this review article could be potentially applied to the study of neuronal migration in various brain regions, but we will specifically discuss examples that have been successfully applied to study migration of cortical neurons. Conventional and conditional deletion of gene function, combined with genetic fate mapping in transgenic mouse models has strongly contributed to our understanding of the routes of migration and the molecules involved in the regulation of this process. However, since this review focuses on the technical details of the methodological approaches to investigate migration, we will not describe transgenic models here and we refer the reader to other reviews [24,25]. In our review, we will divide the approaches to study cortical neuronal migration in in vitro, ex vivo and in vivo techniques and for each technique we will describe the experimental protocols, discuss the applications, the advantages and the main limitations.

### 3.1. In Vitro Assays

In vitro methods are powerful tools to study migration at the cellular, biochemical and molecular levels. These methods provide faster and more reproducible responses on the functional role of genes and proteins implicated in neuronal migration compared to ex vivo and in vivo methods. These methods are also suitable to study intracellular signaling pathways related to cell migration such as cytoskeleton remodeling and cell-matrix adhesion properties [26], especially when combined with live imaging.

Here, we will focus on in vitro migration assays that are useful to quantify the chemotropic migratory response of isolated cortical neurons to specific stimuli. These include the use of transwell-based chemotactic assays and the stripe assay.

In vitro assays to study cortical neuronal migration are based on the use of primary dissociated neurons as a cell source. These techniques have been applied mainly to study the migration of interneurons, which can be easily obtained and isolated by enzymatic and mechanical dissociation of the MGE, from E13.5 [27].

#### 3.1.1. Boyden Chamber/Transwell Assays

##### Technical Details

The Boyden chamber assay is one of the most common in vitro tools used to assess migration of primary neurons. Specifically, this assay allows evaluating the capacity of isolated cells to migrate through a semi-permeable polycarbonate membrane towards a concentration gradient of a soluble (chemotaxis) or substrate-bound (haptotaxis) substance. The substance can be either a chemoattractant or a chemorepellent and both responses can be measured with this method.

Boyden Chamber and more generally transwell migration assays are based on the presence of two different compartments, separated by a microporated membrane (Figure 2A). In the chemotaxis experiments, the lower compartment is filled with the substance of interest (recombinant molecule or conditioned medium from COS cells expressing the cue) while the wells in the upper part are filled with cells, resuspended at specific densities. In the haptotaxis experiments the lower side of the filter is pre-coated with a non-soluble substrate and the lower wells are filled with serum-free medium, while the upper wells with the cells.

When using primary dissociated neurons as a cell-source, the membrane is normally pre-coated with a mix of poly-lysine and laminin to improve adhesiveness of the cells. The chamber or transwells are then incubated at 37 °C in a 5% CO_2_ incubator for a period of time that might differ depending on the characteristics of the cells. For example, primary embryonic neurons derived from MGE are allowed to migrate for 8–16 h. Migrated cells that adhere to the lower side of the filter are then post-fixed and stained to allow cell counts and quantification.

##### Applications

The Boyden Chamber assay has been successfully employed to study the effects of single guidance cues on interneuron migration or to dissect the role of specific signaling pathways. For example, Rakic and colleagues beautifully demonstrated that MGE-cells are able to migrate towards three ErbB4 ligands, which are extensively expressed in vivo by non-interneurons of the developing forebrain [28].

With the same technique, Hernandez-Miranda and colleagues demonstrated that MGE-cells from Robo1-/- mice are not responsive to Semaphorin (Sema) 3A and Sema3F, which are normally implicated in the striatal repulsion of cortical neurons, thus providing evidence of a novel semaphorin-receptor signaling pathway [29].

#### 3.1.2. Stripe Choice Assay

Another in vitro technique to assess migration of cortical neurons is the stripe assay, which measures the differential migration of primary neurons on stripe of COS cells expressing different molecules (Figure 2B). COS cells can be transfected with constructs codifying for chemoattractant or chemorepellent molecules and interneuron migration can be evaluated as the distribution of MGE-derived cells in each stripe. Specifically, this technique has been used to evaluate the effect of molecules such as neuregulins. In these experiments, MGE derived interneurons are plated on a monolayer of COS cell stripes expressing the chemoattractant neuregulin-1 alternated with mock-transfected COS cells and it was observed that interneurons preferentially migrate towards stripe expressing the chemoattractive source [30].

### 3.2. Ex Vivo Assays

In vitro methods based on dissociated neurons represent a powerful method to study distinct neuronal populations and allow morphological, toxicological and functional studies, among which migration assays. However, the lack of cell-cell contacts, well-defined tissue architecture and factors provided by the microenvironment may alter neurons behavior and represent the main limit of in vitro techniques. Moreover, culture conditions, such as extracellular matrix protein exposure or 2D-3D substrates, may alter cell properties (e.g., migration or survival) and gene expression.

To overcome these limitations, over the past years ex vivo techniques have been developed. These include the use of organotypic slices [31] and explant cultures [32].

#### 3.2.1. Organotypic Slices

Organotypic slices are brain slices of 100–400 µm thickness, which can be kept in culture for several days, thus mimicking in vivo conditions. Among the pioneers in brain organotypic slice cultures, Gahwiler, in early 1980s, developed the roller tube culture based on the air/liquid interface culture that allows long-term survival without alterations in cellular differentiation or organization [33]. Years later, in 1991 Stoppini optimized this technique developing a semiporous membrane culture that is the most commonly used culture method nowadays [34,35,36].

Brain organotypic slices can be used to study various aspects of neuronal development and function, such as neuronal networks formation, synaptic plasticity, electrophysiology, axonal transport, chronic toxicology, gene function and also migration of neuronal precursors [37].

##### Technical Details

Brain organotypic cultures can be obtained both from embryonic and postnatal brains, but in this review we will focus on methods that can be applied to study embryonic cortical neuronal migration.

Briefly, after the sacrifice of a pregnant mother at the desired stage, embryos are collected in a sterile Petri dish containing ice-cold artificial cerebrospinal fluid (ACSF) to minimize tissue degradation. Then, brains are removed and quickly embedded in 3% low melting agarose for vibratome cutting. The optimal thickness of embryonic cerebral cortex slices is between 200 and 300 µm. Slices are then collected in a glass bottom multiwell plate on a floating semiporous membrane and grown in a specific culture medium at 37 °C in 5% CO_2_ incubator [38]. Slices can be studied shortly after dissection for acute experiments or grown for several days for long-term experiments by daily media changes.

##### Applications

Although organotypic slices can be used to study the migration of both projection neurons and interneurons, here we will focus on the applications for studying interneuron migration, as other techniques, such as in utero electroporation, are preferred for projection neurons due to technical difficulties in targeting MGE in vivo.

The use of organotypic slices offers the advantage to live monitor the migration of neurons, which can be labeled with vital dyes or reporter genes. In addition, it is possible to evaluate the effects of guidance cues on migration by co-culturing exogenous sources of cues such as aggregates of COS-expressing cells. Further, slices are a good model to study gene function either by generating slices from specific knockout mice or after focal or ex vivo electroporation of short hairpin RNA (shRNA) within MGE.

- Live imaging of interneuron migration

Cortical interneuron migration has been discovered by injecting fluorescent dyes (i.e., carbocyanine dyes such as Dil) into the MGE of organotypic slices [9,39,40,41], which represents one of the methods still used to label specific neuronal populations. Carbocyanine crystals are placed in the slice MGE and uptaken by neurons migrating out of the MGE. The slice containing labeled neurons can be post-fixed and migratory route of labeled interneurons analyzed under a confocal microscope. More recently, several reporter animals have been developed to label specific neuronal population with endogenous fluorescence; the most commonly used mouse reporter lines for interneuron detection are Dlx5/6-Cre-IRES-EGFP [42] and GAD67-GFP [43]. Slices derived from these reporter lines can be employed to live image migrating interneurons and study morphological changes occurring in neurons during the migratory process. Indeed, interneuron migration is made up of different cyclical phases of leading process extension and nuclear movement that involve cytoskeleton remodeling and organelles migration [44,45]. The dynamics of these events can be easily visualized by using such reporter genes.

- Co-culture experiments on slices

Organotypic slices can also be employed to perform co-culture experiments. Using this system, migration of interneurons towards a chemorepellent or chemoattractant can be studied by looking at interneuron trajectories after placing aggregates of COS cells expressing the soluble cue in ectopic regions. For example, COS expressing neuregulin-1 placed in the ventral forebrain of organotypic slices are able to attract interneurons far away from their physiological migratory route by inducing ectopic migration [30].

Rodent cortical organotypic slices also enable to study the fate of heterotypic cells such as of human neural progenitors. In particular, neuronal precursors derived from human embryonic stem cells (hESCs) well integrate into murine cortical slice allowing the study of migratory properties of human-derived cortical neurons. In addition, it is possible to test how several genes (e.g., DCX) affect migration of neurons derived from hESC by transfecting human neurospheres with the gene of interest [46].

Moreover, gain and loss of gene function studies can be performed in slices after focal electroporation of overexpressing vectors or shRNAs for the gene of interest. Such studies revealed the role of several genes in cortical neuronal migration [47,48,49], as also explained in the next section.

- Focal electroporation

The identification of mechanisms regulating tangential migration is mainly based on gain of function (GOF) or loss of function (LOF) studies that can be performed by applying focal electroporation on brain organotypic slices (Figure 3A). This technique consists in the delivery of plasmid DNA into neural progenitors of the MGE. To selectively target interneurons, the injection should be performed very close to the site of their exit from ganglionic eminence beneath the corticostriatal layer. Electrodes are positioned above (vertically oriented negative pole) and at the bottom (horizontally oriented positive pole) of the slice as illustrated in Figure 3A. The electroporation should be performed with extra care, because even minor damages could alter migration [38]. As for the other organotypic slices, the electroporated slices are then kept in culture on floating membranes.

In the majority of the experiments, focal electroporation is used to perform gene knock-down and offers a valid alternative to study the function of genes causing early embryonic lethality in knockout mouse models. The delivered plasmid usually encodes for short hairpin RNA (shRNA) or for non-functional protein (e.g., the dominant negative form of a protein) [30,47,48]; to confirm the specificity of the defects observed after RNA interference, rescue experiments are normally performed by co-transfecting shRNA and constructs codifying for the inactivated gene, as it was done by Frioucourt and co-authors to study the role of DCX gene in interneuron migration [49]. Rescue experiments also provide an efficient tool to distinguish cell-autonomous from non cell-autonomous mechanisms [33]. An alternative method to assess cell-autonomous effects is based on graft experiments combined with organotypic slices: for instance, MGE-derived GFP-positive neurons are transplanted on WT or mutant organotypic slices in order to track perturbations in their migration [50].

In addition, focal electroporation can be used to overexpress genes in gain-of-function studies, as shown by Van den Berghe and co-authors for the Unc5b gene. In these experiments, the overexpression of the guidance receptor Unc5b in interneurons disrupted their migration [51].

Differences in interneuron migration between wild-type or mock-transfected slices versus slices transfected with an overexpressing or shRNA vector are usually assessed by measuring the distance travelled by neurons from the MGE or by quantifying the number/proportion of electroporated cells in the cortex [49]. As expected, to obtain a reliable quantification many replicates need to be analyzed as this technique provides highly variable results. Indeed, the success of focal electroporation strictly depends on technical aspects such as the correct injection site and the transfection efficiency.

#### 3.2.2. Ex Vivo Explants

The study of interneuron migration in organotypic slices is complex mainly because the quantification of migration is hard to perform on slices, since interneurons are spread throughout the whole forebrain. Thus, MGE explants are an alternative reliable tool to quantify the morphological changes that occur during migration and to measure parameters such as speed and directionality [44,52]. Dissected MGE are usually cultured above a monolayer of homochronic wild-type mixed cortical cells (feeder layer) to mimic in vivo conditions. Alternatively, it is possible to culture MGE explants in matrigel together with an attractive or repulsive source (Figure 3B). The source can be a cortical explant that provides long-range chemotactic factors [30,53] or a recombinant molecule (e.g., BDNF) [54] or an aggregate of cells producing molecules that regulate migration (e.g., COS cells expressing neuregulin 1, neuregulin 3, CXCL12) [30,55,56,57]. Thus, the effects of the chemotropic source on interneuron migration from MGE can be quantified by dividing the space around the explants in quadrants and by counting the number of cells in each quadrant. In addition, explants can be used to measure velocity and maximum distance travelled by interneurons, when combined with live imaging [51].

Recently, a new method that allows the analysis of individual neurons migrating out of the MGE explant towards the cortex has been developed. This method is based on a microfluidic device, which is composed by two compartments filled with matrigel. One MGE and cortical explant is placed in each compartment and the migration of single interneurons towards the cortex can be visualized in microchannels that are connecting the two compartments [53]. Because this technique allows focusing on single migrating neurons, the quantification of migration is simplified. In addition, this technique allows studying intracellular dynamics such as organelles re-arrangements, thus providing insights into both cellular and molecular mechanisms.

### 3.3. In Utero Electroporation to Study In Vivo Cortical Neuronal Migration

Neuronal migration in the developing cortex can be studied directly in vivo in embryos using the technique of in utero electroporation [58]. This approach allows visualization of fluorescently labeled migrating neurons and their genetic manipulation in the native environment.

Before the advent of in utero electroporation, cortical neuronal migration has been studied by looking at the position of labeled neurons in birth-dating experiments. In the 1970s, pioneering work by Pasko Rakic set the ground for the study of cortical projection neuronal migration in mammals. Rakic injected pregnant monkeys with tritiated thymidine, a radioactive labeled DNA base analog, which gets incorporated into dividing neural progenitor cells. When a progenitor exits the cell cycle and differentiates, the newly generated neuron will remain permanently labeled and it can be traced in the adult brain.

In this way Rakic revealed that tritiated positive neurons were located far from the proliferative areas, in distinct positions across the cortical plate. Moreover, he showed that neuronal positioning was not random but followed a specific pattern, known today as “inside-out”, which is characterized by early born cells located in deeper cortical layers and later born neurons populating more superficial regions [59]. Using high-resolution electron microscopy, Rakic was also able to show that the nuclei of migrating neurons were juxtaposed to radial glia fibers, the earliest indication of radial-guided migration [60]. Although the basic model of cortical development and neuronal migration proposed by Rakic was correct, it is not until two decades later that details of radial and tangential migration started to be discovered with the introduction of in utero electroporation.

In 1997, Muramatsu and colleagues showed for the first time that electroporation of plasmids in ovo in the chicken neural tube was the most powerful non-viral technique to mark neural cells [61]. In 2001, various laboratories around the world have successfully applied the electroporation technique to mark and trace newborn cortical neurons in the embryonic cortex of mice and rats [62,63,64]. Such studies revealed that radial migration is a multistep process characterized by sudden changes in cell morphology, polarity and speed as neurons travel from the VZ to the CP (see details in applications-cellular dynamics and morphological changes). Moreover, subsequent adaptation of the technique enabled insights into tangential migration in vivo to be gained for the first time [65].

#### 3.3.1. Technical Details

In utero electroporation consists in the delivery of plasmid DNA into neural progenitors in live brain embryos [66,67]. This is achieved by performing regulated surgical procedures on anesthetized pregnant mice. During this procedure, the entire uteri are exposed and the embryos carefully oriented with their heads and brains visible from the transparent amniotic membrane (Figure 4A). The DNA of interest is injected into the lateral ventricles of the embryonic brain using a fine glass capillary with a sharp and thin tip. Injection and diffusion of the DNA into the ventricular cavities can be guided by co-injection of a dye, such as FastGreen. The DNA is then electroporated into the cells that are lining the lateral ventricles, by application of pulses of currents through the embryo head. Since DNA is negatively charged, it will follow the direction of the positive electrodes. Therefore, the position of the electrodes will ultimately determine the targeted site (Figure 4B). Targeting of radially migrating cortical neurons is achieved by positioning the positive electrode on the dorsolateral side of the brain, whereas targeting of ventrally-located progenitors requires 90 °C rotation of the electrodes with the cathode placed at the base of the brain, as shown in Figure 4B.

Moreover, at this stage, it is very important to avoid touching the placenta with the electrodes, because this is one of the major causes of embryo death.

Once the electroporation is terminated, the embryos are gently pushed back into the abdominal cavity and the animal let to recover post surgery. The analysis of migration can be performed any time after electroporation, from a few days later until postnatal stages.

#### 3.3.2. Applications

- Quantification of migration

Most of the studies on mammalian cortical neuronal migration have been performed in the mouse. The timing of cerebral cortex development in this model is particularly advantageous, since neurogenesis starts around the embryonic day E10 and proceed till birth 10–12 days later [6,68]. Therefore, cells undergo considerable migration in only a few days. Visualization of radial migration can be achieved by targeting dorsally-located cortical progenitors with a construct expressing the Green Fluorescent Protein GFP or other fluorescent proteins, at a specific embryonic stage. As development progresses, GFP-labeled cells will differentiate and migrate to the cortical plate. Analysis of the cortices 3 days after electroporation shows that GFP positive cells have reached the upper part of the cortical wall (Figure 4C). This process can be quantified, by counting the number of cells in each compartments, e.g., VZ, SVZ, IZ and CP, and by representing it as a percentage over the total number of electroporated cells (Figure 4D). In the example shown in Figure 4C,D, nearly 50% of cells reach the CP 3 days after electroporation, whereas only a minor fraction remains in the deeper SVZ/VZ regions.

In contrast to radial migration, the study of tangentially migrating interneurons has been more challenging for two main reasons: on one hand because of the difficulty in targeting restricted areas of the ventral forebrain (e.g., MGE, LGE) and on the other hand because of the difficulty in quantifying the number of cells reaching and spreading throughout the cortical plate. Such quantification would indeed require whole forebrain analysis due to tangential spreading of neurons over long distances from their birthplace. Thus, the most common technique used to target cortical interneurons is to perform focal electroporation in brain slice culture or ex vivo explants, as described in Section 3.2.1.

- Cellular dynamics and morphological changes

Pioneer work from O’Rourke and colleagues using time-lapse imaging in cortical slices allowed visualization of the morphological changes that cortical neurons undergo during their migration [69]. Through in utero electroporation, combined with live imaging, it is now possible to investigate such morphological dynamics in vivo. According to changes in neuronal shape and speed of migration, the process of radial migration can be divided in distinct phases (Figure 5).
(1)The initial step of radial migration consists in the detachment of the apical process from the VZ surface, which is accompanied by a loss of bi-polarity by RGCs (Figure 5-phase 1). Deregulation of apical detachment disrupts the integrity of the neural epithelium and results in neuronal migration disorders as shown for example by perturbing FilaminA and β1 integrin function [70,71].(2)Post-mitotic cells leave the proliferative VZ and SVZ and reach the IZ, where they acquire a multipolar shape, which is characterized by multiple protrusions that dynamically extend and retract from the cell body (Figure 5-phase 2). Several genes have been shown to play important roles in exiting the IZ, since their knock down causes accumulation of multipolar cells in this region. Interestingly, the majority of these genes are involved in cytoskeletal remodeling via the actin or microtubule networks, such as ENA/VASP homology proteins (EVH), Dcx, Lis and Rnd2 [72,73,74].(3)After leaving the IZ, cells acquire a bipolar shape, with a leading process facing the pial surface and a trailing process facing the VZ. The leading process of migrating neurons embraces the radial glia filament and uses it as a guide for migration. Bipolar cells migrating in the CP undergo a mode of radial migration called locomotion (Figure 5-phase 3). This is characterized by repetitive cycles of leading process extension, forward displacement of the nucleus and partial retraction of the trailing process. These events require strict coordination between centrosomal and nuclear movements, as shown by several studies that use fluorescently labeled centrin to track centrosome in in vivo migrating cells [75,76].(4)Once neurons reach the last part of the CP, they attach the leading process to the outer brain surface and translocate their nuclei upwards, using another mode of radial migration, called somal translocation (Figure 5-phase 4) [77].

While radial migration has been well characterized, less is known about the cellular dynamics of tangentially migrating cortical interneurons in vivo. The use of multiphoton and two-photon microscopy has been successfully used to image migrating interneurons in live embryos [20,78]. These studies revealed that migrating interneurons undergo multidirectional movements once they invade the cortex from the ganglionic eminence [78]. By observing migrating interneurons in the superficial marginal zone, these works also showed that interneurons first disperse tangentially, then extend processes towards the cortical layers beneath and finally dive in to integrate into specific cortical laminae [20]. In vivo imaging of interneurons thus provides a powerful way to gain insights into interneuron migration in vivo, however such approach is technically challenging and limited in the duration of the time-lapse recording. Therefore, detailed studies on interneuron migration have been mainly performed in brain slices. For example, ex vivo studies using focal electroporation in brain slices to mark MGE-derived GABAergic interneurons revealed that the leading process of interneurons does not follow specific scaffolds, but rather branches multiple times before choosing a direction [79]. The highly dynamic branching of the leading process, together with the in vivo observations that interneurons undergo multidirectional movements before choosing a specific direction, likely represents a mechanism to explore the extracellular environment, as indeed migration of GABAergic interneurons is known to strongly rely on extrinsic signals [3,80].

- Gene function studies

In utero electroporation is a rapid and efficient method to perform gene functional studies. Overexpression or silencing of a gene of interest can be achieved by electroporation of the gene coding sequence or a gene-specific short hairpin RNA (shRNA), respectively. Co-expression of a fluorescent protein, such as GFP, from either the same overexpression or shRNA construct or by co-electroporation of a separate GFP-expressing vector is used to visualize and track the position of the electroporated cells. Importantly, when co-electroporating two different plasmid vectors, almost 100% of electroporated cells contain both plasmids. By changing the promoter driving gene expression, we can also spatiotemporally control the dynamics of gene expression. For example, the use of neuronal specific promoters, such as Nex, Dcx or NeuroD1 promoters, has been used to drive gene expression only in post-mitotic neurons [72]. This allows clear resolution between the effects that a gene has on the early steps of neurogenesis versus the later stages of post-mitotic neuronal migration. Moreover, temporal control over transgene expression can be further refined by using a tamoxifen inducible form of the Cre recombinase (CreERT) in a Cre/loxP-mediated inducible expression system [81].

The promoter generally used to drive shRNA expression is the U6 for RNA polII transcription. In order to achieve inducible gene silencing, an elegant strategy has been developed, which uses a Cre-Lox based conditional shRNA vector. This vector carries a stop cassette flanked by loxP sites and, when injected for example in NexCre mice, neural specific removal of the stop cassette allows shRNA expression only in post-mitotic neurons. This strategy has been used for example to study the role of Rnd3 in postmitotic neurons [75].

One of the best characterized genes to be silenced in the rodent cerebral cortex has been Doublecortin (Dcx). Dcx is part of a small family of microtubule-associated proteins that control the stability and polymerization of the MT network [82]. Human mutations in DCX gene cause X-linked lissencephaly (smooth brain) in males and doublecortex syndrome in females [73]. Genetic knockdown of Dcx in the developing rat neocortex results in abnormal radial migration with effects on both the multipolar to bipolar transition in the IZ [83] and the nucleus-centrosome uncoupling during locomotion in the CP in the mouse [84]. It is important to note that Dcx knockout mice do not show overt migration abnormalities [85,86], thus highlighting the importance of acute knockdown via in utero electroporation to overcome compensation and study gene function.

Altogether, gene function studies performed via in utero electroporation led to the discovery of novel genes implicated in the aetiology of neurodevelopmental disorders (for examples, refer to [13,87,88,89,90]) and provide an experimental platform to investigate genetic interactions and molecular mechanisms underpinning the cellular activities of such genes.

- CRISPR/Cas9

In recent years, novel genome editing technologies have been developed to facilitate in vivo genetic manipulations. These approaches rely on the genetic deletion or insertion via homology directed repair (HDR) or non homologous end joining (NHEJ) following injection of nucleases of the ZNF, TALENs or CRISPR/Cas9 classes, which provoke site-specific double strand DNA breaks [91,92,93]. In particular, CRISPR/Cas9 technology has become the nuclease system of choice, mainly because it is fast, efficient and cost-effective. CRISPR/Cas9 originates from bacterial adaptive immune system and functions by directing the Cas9 nuclease to the genomic locus complementary to the sequence of the guide RNA (gRNA). In utero electroporation of gRNA and Cas9 has been recently used to knock down and knock in genes in cortical progenitors. Kalebic and colleagues successfully used gRNA to knock down the transcription factor Tbr2 in neural progenitors and showed that loss of Tbr2 generates neurogenesis defects [94]. Tnsunekawa and colleagues further implemented the technique to insert genes in specific genomic loci and to control for zygosity [95]. In order to perform Crispr mediated and locus specific knock in of fluorescent proteins, the authors co-electroporated the targeting vector carrying eGFP or RFP with gRNA and Cas9, which mediates DNA double strand break (DSB)-induced HDR with the targeted plasmid sequence. Via this approach 10% of cells are targeted homozygously, making this approach a fast and direct way to visualize and analyze homozygous acute gene knock out cells.

In the future, genetic manipulation in utero via CRISPR/Cas9 technology is likely to flourish, however it is important to remember that at this time, it is still at its infancy and several controls, including Cas9-negative control and sequencing of amplicons containing potential off-target loci, are necessary to assess the functionality and the specificity of the system.

- Signalling pathway studies (RhoGTPases, Calcium indicator)

The power of in utero electroporation goes beyond studying the function of a single gene. Indeed, this technique provides the possibility to decipher in vivo the signaling pathways that control neuronal migration dynamics in fine detail. For example, the state of activation of RhoGTPases, which controls cytoskeletal remodeling and cellular adhesion, can be investigated directly in situ, by electroporation of intramolecular FRET (Fluorescence Resonance Energy Transfer) probes for the specific RhoGTPase of interest, such as RhoA [75,96,97,98]. The principle of FRET, whereby energy transfer occurs between two fluorophores in close proximity, has been employed to design intramolecular probes, such as the one for RhoA [99]. In this probe construct, RhoA, conjugated to one fluorophore (CFP), is also fused to the rhotekin-binding domain (RBD), which in turn is conjugated to a different fluorophore (YFP). Activation of RhoA determines binding to RBD through an intramolecular conformational change that brings the two fluorophores close to each other, thus increasing FRET efficiency signal. Variations in FRET efficiency therefore reflect differential RhoA activation and can be used to track RhoA variations in different cellular compartments or during distinct phases of neuronal migration. For example, it has been shown that a decrease in RhoA activity is necessary for the initiation of cortical neuronal radial migration [75].

Another biosensor that can be introduced via in utero electroporation measures the levels of intracellular Calcium [100,101]. GCaMP is a genetically encoded calcium indicator, which exhibits bright fluorescence when elevated Calcium levels are present in the cells and bound to the GFP-fused protein Calmodulin. Experiments performed using this Calcium indicator revealed that Calcium waves are important to regulate not only cortical neuronal migration, but also neurogenesis by providing feedback information along the radial glia migratory pathway.

## 4. Advantages and Limitations of the Techniques

Altogether the experimental approaches here described provide useful tools to study the mechanisms regulating cortical neuronal migration. In vitro models offer a fast, straightforward and low-cost solution to quantitatively study neuronal migration. However, cells are analyzed out of their native environment and therefore results should be interpreted with caution, as primary cells have to migrate in the presence of limited extracellular matrix components and in the absence of three-dimension tissue architecture. In addition, in vitro assays, such as transwell assays, do not allow tracking cell migration in real time, therefore limiting the possibility to observe dynamic changes in the migration and/or morphology of the cells. Nevertheless, these assays are extremely useful to assess the specific effect of single molecular cues without the influence of other factors and to confirm the results obtained with in vivo approaches.

Ex vivo techniques, such as brain organotypic slices, offer methods that overcome these limitations and represent a good compromise between in vitro and in vivo approaches. Organotypic slices well mimic the in vivo conditions but at the same time allow an easier tissue manipulation compared to living animals. Further, organotypic slices represent the choice method to study interneuron migration, as MGE is easy to visualize and access in these preparations. However, organotypic slices also have some limitations. For example, they cannot be kept in culture for a long period of time, thus limiting the analysis of migrating neurons to relatively short time windows. Moreover, quantification of migrating neurons in organotypic slices can be highly variable, due to both operator and technical differences (e.g., the injection site in focal electroporation should be always the same) and therefore a high number of samples should be analyzed to confirm the results.

The introduction of in utero electroporation contributed to expedite our understanding of the cellular and molecular mechanisms of brain development and cortical neuronal migration. Gain and loss of gene function studies performed via in utero electroporation have the great advantage of being rapid and efficient methods in comparison to time-consuming knockout and knockin strategies. Importantly, the advent of Crispr/Cas9 technology is accelerating the time frame of generating transgenic mice and will therefore become an essential complement to in utero electroporation results. Another advantage of performing genetic studies via in utero electroporation is that spatial and temporal control of the targeted cell type can be achieved by using cell type specific promoters and by controlling the direction of the electric current. The possibility to target distinct areas depending on the position of the electrodes is a great advantage for example over viral infections where the infected area cannot be controlled. However, there are still limitations in the degree of specificity that can be achieved by simply rotating the electrodes. It is for example still challenging to target distinct ventral subdomains, which are contiguous to each other, but give rise to different types of striatal and cortical interneurons. In order to increase the precision of the targeted site, the use of triple electrodes has recently reported impressive results [102,103]. Other types of neurons that are difficult to target with electroporation are the earliest born neurons, mainly because of the size of the embryos at these early stages. Therefore, ex vivo electroporation followed by whole embryo culture is generally preferred [104].

In utero electroporation is an extremely versatile approach to study not only radial migration but also the earlier and later neurogenic events, such as neural progenitor proliferation and differentiation and the formation of axons, dendrites and synapses. However, due to the low number of electroporated cells, in depth molecular characterization (transcriptomic, proteomic) of GOF and LOF phenotypes has been so far limited. In the future, as transcriptomic and sequencing technologies advance to accommodate the analysis of a low number of cells, the range of applications of in utero electroporation will also grow.

## 5. Conclusions

In conclusion, this review describes the main modes of neuronal migration in the developing mammalian cerebral cortex and summarizes the current techniques to study this process. Deciphering the molecular mechanisms that control cortical neuronal migration is of invaluable help to understand the etiology of human disorders, which are due to defective neuronal migration. The combination of next-generation sequencing and genome-editing tools will further increase the applications of the methods here described and will enhance our knowledge on the subject.

## Figures and Tables

**Figure 1 brainsci-07-00048-f001:**
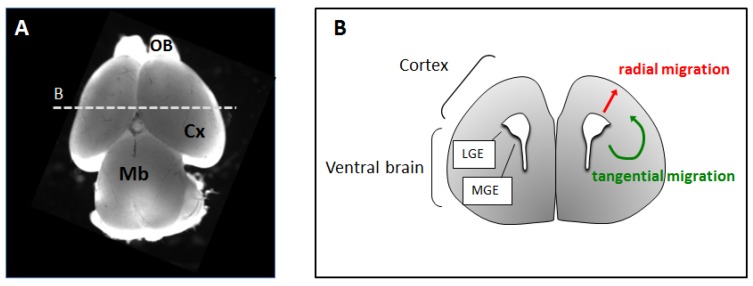
(**A**) Top view of a mouse brain at embryonic stage E15.5. OB: olfactory bulb, Cx: Cortex; Mb: Midbrain. (**B**) Schematic representation of a coronal section of a mouse embryonic brain through the plane B. Routes of radial and tangential neuronal migration are shown in red and green, respectively. LGE: lateral ganglionic eminence; MGE: medial ganglionic eminence

**Figure 2 brainsci-07-00048-f002:**
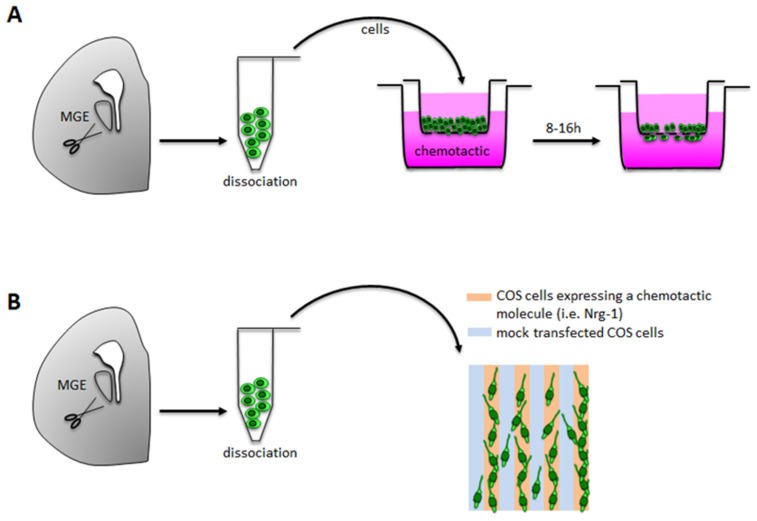
In vitro techniques to study interneuron migration. Primary dissociated interneurons are obtained from MGE explants by enzymatic and mechanical dissociation. (**A**) Transwell/Boyden chamber assays are based on an upper compartment (light pink) containing primary neurons and a lower compartments containing the chemotactic molecule creating a gradient from the bottom to the top of the well (dark pink); compartments are separated by a porous membrane through which neurons migrate; migration rate is evaluated after 8–16 h of incubation. (**B**) Stripe choice assay is based on monolayers of transfected (light orange) and mock transfected (light blue) COS cells; primary interneurons are plated and differentially migrate on stripes, with a preference for stripes of transfected COS cells if expressing a chemoattractant.

**Figure 3 brainsci-07-00048-f003:**
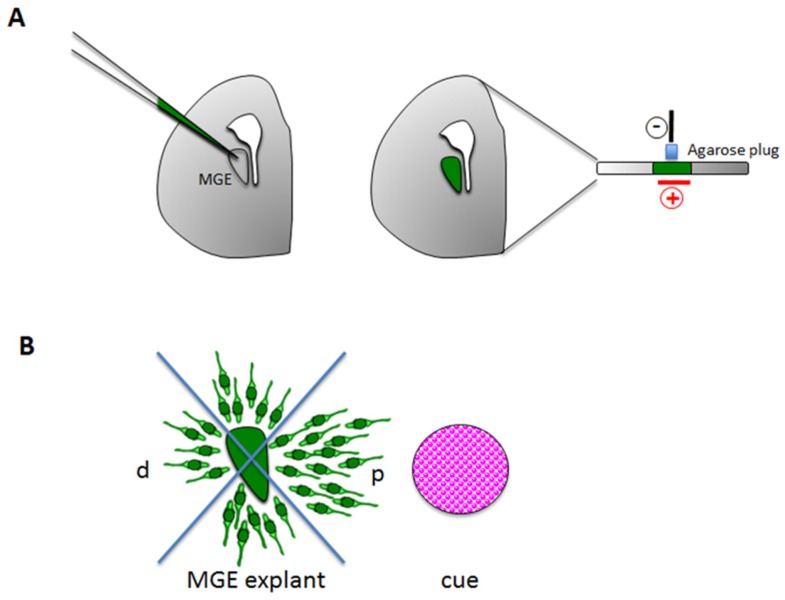
Ex vivo techniques to study interneuron migration. (**A**) Schematic representation of focal MGE electroporation: MGE is injected with a DNA vector or RNA (e.g., shRNA) and then electroporated to allow the DNA/RNA to enter interneurons. An agarose plug is used to separate the electrode from the slice. (**B**) MGE explants can be co-cultured in the presence of a chemoattract/chemorepellent cue, which could be a recombinant protein, an aggregate of protein–expressing cells or another tissue explant such as cortical explant. Migration is quantified by counting neurons in proximal (p) versus distal (d) quadrants

**Figure 4 brainsci-07-00048-f004:**
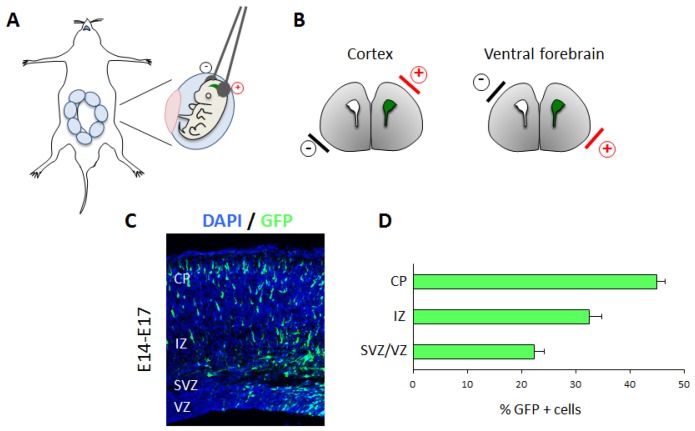
(**A**,**B**) Schematic representation of in utero electroporation. Uteri are exposed from the abdominal cavity and embryonic brains are electroporated. The position of the electrodes determines the electroporated area, as indicated in (**B**). (**C**) Representative results of radial migration analysis. Section of E17 embryonic cortex electroporated with GFP vector at E14. Nuclei are counterstained in blue. (**D**) Quantification graph showing the distribution of GFP-positive cells in different zones of the cortex (VZ/SVZ, IZ and CP). The majority of GFP positive cells have migrated to the cortical plate. VZ: ventricular zone; SVZ: subventricular zone; IZ: intermediate zone; CP: cortical plate.

**Figure 5 brainsci-07-00048-f005:**
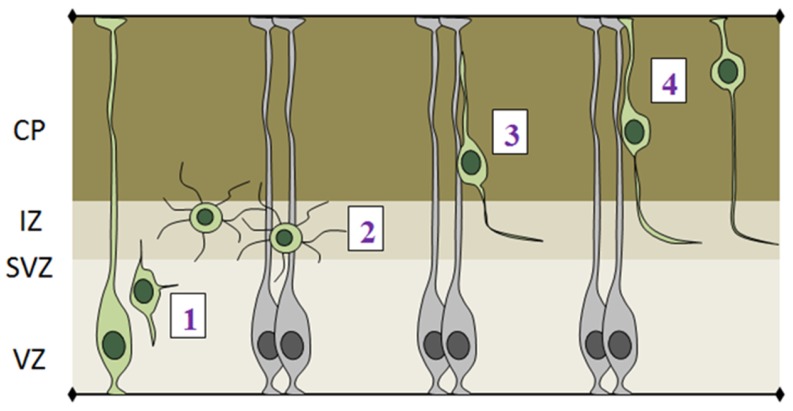
Phases of radial migration of cortical projection neurons. Newborn neurons leave the proliferative areas (1) and reach the intermediate zone where they exhibit a multipolar morphology (2). Neurons exiting the intermediate zone acquire a bipolar morphology and migrate through locomotion using radial glia fibers as scaffold (3). When neurons reach the basal surface they attach the leading process to the pial surface and translocate only the nucleus (4). VZ: ventricular zone; SVZ: subventricular zone; IZ: intermediate zone; CP: cortical plate.

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
