# Peer review of "In Vitro, Ex Vivo and In Vivo Techniques to Study Neuronal Migration in the Developing Cerebral Cortex"

_brainsci, 2017, doi:10.3390/brainsci7050048_

Round 1
Reviewer 1 Report
This is a timely, well-written review of the topic. It is well introduced and comprehensively reviews the various approaches used to study cortical neuronal migration. The authors cover the pros and cons of the techniques, how they have been used to quantitate migration, to understand the cellular dynamics and morphological changes involved, as well as their use in gene function and signaling studies. The figures clearly illustrate and support the points raised in the text. My suggestions are minor and grammatical.
1. Some words are unintentionally hyphenated e.g. lines 70, 99
2. Line 219 replace “cyclic different” with “different cycling”
3. Line 247 replace “Like” with “As”
4. Line 319, replace “for the first time to gain insights into tangential migration in vivo” with “insight into tangential migration in vivo for the first time”
5. Line 372, replace “in” with “into”
6. Line 467, replace “reflects” with “reflect”
Author Response
We thank the reviewer for all his/her constructive comments. Please find detailed responses to each comment below:
This is a timely, well-written review of the topic. It is well introduced and comprehensively reviews the various approaches used to study cortical neuronal migration. The authors cover the pros and cons of the techniques, how they have been used to quantitate migration, to understand the cellular dynamics and morphological changes involved, as well as their use in gene function and signaling studies. The figures clearly illustrate and support the points raised in the text. My suggestions are minor and grammatical.
1. Some words are unintentionally hyphenated e.g. lines 70, 99
2. Line 219 replace “cyclic different” with “different cycling”
3. Line 247 replace “Like” with “As”
4. Line 319, replace “for the first time to gain insights into tangential migration in vivo” with “insight into tangential migration in vivo for the first time”
5. Line 372, replace “in” with “into”
6. Line 467, replace “reflects” with “reflect”
Reply:
We thank the reviewer and edited the text accordingly.
Reviewer 2 Report
Comments to Authors:
Manuscript ID brainsci-188619
Authors: Azzarelli et al.,
“In-vitro, ex-vivo and in-vivo techniques to study neuronal migration in the developing cerebral cortex”
The manuscript “In-vitro, ex-vivo and in-vivo techniques to study neuronal migration in the developing cerebral cortex” submitted by Azzarelli and others, provides an informative, concise, up to date and, on the whole well-written review of the current techniques employed to study neuronal migration in the embryonic murine forebrain. The figures are clear and well-illustrated. The manuscript will be of interest to readers in the fields of cortical development and neuronal migration.
The authors categorise methodologies into three major sections, based on whether analysis of neuronal migration is carried out using: 1) isolated primary cells in vitro, 2) ex-vivo cortical explants or organotypic slice cultures or 3) in-vivo techniques such as in-utero electroporation and gene knock-down techniques. The authors clearly explain all techniques as well as their advantages and limitations.
There are only a few points the authors may wish to consider, namely, certain sections could be expanded on, as detailed below; as well as some minor stylistic comments regarding the writing of the manuscript.
Comments:
Section 2.2. Ex-vivo assays : Lines 170-174:
The authors could provide more details on the limitations of using in-vitro techniques to study neuronal migration, by mentioning studies which have shown that culturing cells exposed to different extra-cellular matrix proteins, as well studies showing that 2-D vs 3-D substrates can alter gene-expression and migratory properties of cells .
Section 2.1 :In-vitro assays:
The authors only consider/elaborate on the study of chemotropic guidance cues, however in-vitro assays are also useful to study intra-cellular, cellular-substrate interactions as well as intracellular-signalling activities when combined with live-imaging. This section could briefly include these other uses if the word-limit permits.
Section 2.2.2 :Ex-vivo assays: Lines 286-291:
The micro-fluidic assay to investigate single cell migration is one of the more recent techniques described in the review, however the authors do not specify why being able to study individually migrating neurons may be useful or the significance of this technique.
Section 2.3 :In-utero electroporation to study in-vivo migration:
The authors nicely describe the advantages and uses of in-utero electroporation to study migrating neurons in their native environment in this section, however, they should also add a section to include the (albeit technically challenging) use of multi-photon and two-photon microscopy techniques which have been used to image migrating neurons in living embryos (eg. Ang et al., 2003; Yokota et al., 2007) . This is the most important section to expand.
Minor comments:
Introduction :
Line 18: ‘along’ various routes
Line 18: positions
Line 27: to their final positions
Line 38: neuronal migration in mouse models
Lines 40-72:
While the authors nicely introduce the cerebral cortex in humans (lines 40-43), they next mention the use of rodent models to study two major modes of radial and tangential migration in the cerebral cortex without emphasising the similarities (and differences) during rodent and human brain development. They may want to provide a brief sentence here which would link this more smoothly with lines 61-72 in which they emphasise how neuronal migration disorders underlie neurodevelopmental disorders in humans. The authors may also wish to distinguish neurodevelopmental disorders due to defects in radial migration (described in lines 61-72) vs disorders linked to tangential migration defects (eg epilepsy, schizophrenia, autism).
Line 43: through the coordinated processes
Line 48: glutatmate as their
Line 49: the ventricular
Line 70: common (remove hyphen)
Lines73-74:
A linker sentence here re-emphasising the two phases of migration may be useful here.
Line 83: migrate past the earlier born neurons
Line 83: progressively populate
Line 85: six-layered organization
Line 91: the POA also gives rise to 10% of interneurons (Gelman et al., 2009, 2011)
Line 101: glial process
Lines 105-106:
The authors may want to add a linker sentence here in which they emphasise the significance of studying neuronal migration before introducing this section.
Line 118: why only at E13.5? , from E13.5?
Line 135: mechanical dissociation
Line 143: When using primary dissociated neurons as a cell-source, the membranes..
Line 146: MGE are allowed to migrate
Line 148: counts
Line 151: not clear what authors are distinguishing here or how testing interneurons response to specific guidance cues vs ‘dissecting the role of specific signalling pathways’ can be addressed in a boyden assay
Line 198: media changes
Line 202: specify here why in utero electroporation is preferred for projection neurons in comparison to interneurons (technical difficulty in targeting MGE)
Line 215: to label specific neuronal populations
Line 219: made up of different cyclical phases
Line 232: cells (hESCs) integrate into murine cortical slices
Line 286: distance travelled by interneurons when combined with live-imaging
Line 289: cortical explant
Line 300: exits the cell cycle
Line 307: electron microscopy
Line 319: insights into tangential migration in vivo to be gained
Line 364-365: not clear why it is challenging to quantify interneurons which have reached the cortical plate – emphasise that they spread tangentially so requiring whole forebrain to be analysed?
Line 369: Through in-utero electroporation combined with live imaging,
Line 396: centrin to track centrosomes
Line 399: and translocate their nuclei upwards
Line 434: nucleus-centrosome uncoupling
Line 438: interactions
Line 444: Introduce abbreviation for double strand DNA break here (DSB)
Lines 458-459: If the word-limit allows, elaborate on the controls which need to be carried out for the Crispr/Cas9 technology .
Line 464: FRET technique is not introduced
Line 483-484: Unclear what is meant by ‘in-vitro assays do not allow tracking of cell migration in real-time’ as the stripe assays can be combined with live-imaging ? Or do you mean that it’s not temporally physiologically-relevant?
Line 501: Should mention that the advent of CRISPR/Cas9 technology is accelerating the time frame of generating transgenic mice
Other small changes throughout manuscript:
Use italics for terms in-vitro, in-vivo
Use adjective plasmid DNA not plasmidic DNA
Use adjective neuronal not neuron
Author Response
We thank the reviewer for all his/her constructive comments. Please find detailed responses to each comment below.
Section 2.2. Ex-vivo assays: Lines 170-174:
The authors could provide more details on the limitations of using in-vitro techniques to study neuronal migration, by mentioning studies which have shown that culturing cells exposed to different extra-cellular matrix proteins, as well studies showing that 2-D vs 3-D substrates can alter gene-expression and migratory properties of cells.
Reply:
We thank the reviewer for this comment and we edited this section accordingly. However, since we haven’t analyzed 2D and 3D in vitro culture we only highlighted how the exposure to extracellular matrix molecules could perturb cell migration.
Specifically, we added the following sentence: “Moreover, culture conditions, such as extracellular matrix protein exposure or 2D-3D substrates, may alter cell properties (e.g. migration or survival) and gene expression” (lines 211-212)
Section 2.1 In-vitro assays:
The authors only consider/elaborate on the study of chemotropic guidance cues, however in-vitro assays are also useful to study intra-cellular, cellular-substrate interactions as well as intracellular-signalling activities when combined with live-imaging. This section could briefly include these other uses if the word-limit permits.
Reply:
We thank the reviewer for this useful comment and we edited accordingly the relevant section. At the start of paragraph 2.1 (lines 142-147), we added: “In vitro methods are powerful tools to study migration at cellular, biochemical and molecular levels. These methods provide faster and more reproducible responses on the functional role of genes and proteins implicated in neuron migration compared to ex vivo and in vivo methods. These methods are also suitable to study intracellular signaling pathways related to cell migration such as cytoskeleton remodeling and cell-matrix adhesion properties, especially when combined with live-imaging. Here, we will focus on in vitro migration assays that are…”
Section 2.2.2 Ex-vivo assays: Lines 286-291:
The micro-fluidic assay to investigate single cell migration is one of the more recent techniques described in the review, however the authors do not specify why being able to study individually migrating neurons may be useful or the significance of this technique.
Reply:
We thank the reviewer for this comment and we edited accordingly the corresponding section. At the end of paragraph 2.2.2 (lines 337-340), we added the following sentence: ”Because this technique allows to focus on single migrating neurons, the quantification of the migration is simplified. In addition, this technique allows to study intracellular dynamics such as organelles re-arrangements, thus providing insights into both cellular and molecular mechanisms.”
Section 2.3 In-utero electroporation to study in-vivo migration:
The authors nicely describe the advantages and uses of in-utero electroporation to study migrating neurons in their native environment in this section, however, they should also add a section to include the (albeit technically challenging) use of multi-photon and two-photon microscopy techniques which have been used to image migrating neurons in living embryos (eg. Ang et al., 2003; Yokota et al., 2007). This is the most important section to expand.
Reply:
We thank the reviewer and we expanded this section accordingly.
See lines 459-468: “The use of multiphoton and two-photon microscopy has been successfully used to image migrating interneurons in live embryos. These studies revealed that migrating interneurons undergo multidirectional movements once they invade the cortex from the ganglionic eminence. By observing migrating interneurons in the superficial marginal zone, these works also showed that interneurons first disperse tangentially, then extend processes towards the cortical layers beneath and finally dive in to integrate into specific cortical laminae. In vivo imaging of interneurons thus provides a powerful way to gain insights into interneuron migration in vivo, however such approach is technically challenging and limited in the duration of the time-lapse recording. Therefore, detailed studies on interneuron migration have been mainly performed in brain slices.”
Minor comments:
Introduction:
Line 18: ‘along’ various routes
Line 18: positions
Line 27: to their final positions
Line 38: neuronal migration in mouse models
Reply:
All these minor comments have been taken into account.
Lines 40-72: While the authors nicely introduce the cerebral cortex in humans (lines 40-43), they next mention the use of rodent models to study two major modes of radial and tangential migration in the cerebral cortex without emphasising the similarities (and differences) during rodent and human brain development. They may want to provide a brief sentence here which would link this more smoothly with lines 61-72 in which they emphasise how neuronal migration disorders underlie neurodevelopmental disorders in humans. The authors may also wish to distinguish neurodevelopmental disorders due to defects in radial migration (described in lines 61-72) vs disorders linked to tangential migration defects (eg epilepsy, schizophrenia, autism).
Reply:
We thank the reviewer and we edited the corresponding section by taking into account his/her useful suggestions.
We added: “Indeed, the basic mechanisms of cortical development and the overall cortical arealization and cellular distribution are conserved between mice and humans and the relatively fast brain development in mice makes them very useful models. However, it is important to note that major differences, such as in size and gyrification, distinguish the human from the rodent brain and thus some principles of neurogenesis might differ.” (lines 46-50)
Line 43: through the coordinated processes
Line 48: glutatmate as their
Line 49: the ventricular
Line 70: common (remove hyphen)
Reply:
We edited these typos.
Lines 73-74: A linker sentence here re-emphasising the two phases of migration may be useful here.
Reply:
We thank the reviewer and added a linker sentence:” Below we will discuss the biology of the two main different modes of migration in the cerebral cortex: radial and tangential migration.” (lines 90-91)
Line 83: migrate past the earlier born neurons
Line 83: progressively populate
Line 85: six-layered organization
Line 91: the POA also gives rise to 10% of interneurons (Gelman et al., 2009, 2011)
Line 101: glial process
Reply:
All these small mistakes have been edited.
Lines 105-106: The authors may want to add a linker sentence here in which they emphasise the significance of studying neuronal migration before introducing this section.
Reply: We thank the reviewer and added a linker sentence: ”Research on the molecular mechanisms regulating cortical neuronal migration is relevant not only to understand brain development, but also to gain insights into the mechanisms that underlie several neurological disorders.” (lines 128-130)
Line 118: why only at E13.5? , from E13.5?
Line 135: mechanical dissociation
Line 143: When using primary dissociated neurons as a cell-source, the membranes..
Line 146: MGE are allowed to migrate
Line 148: counts
Reply:
All these typos have been edited.
Line 151: not clear what authors are distinguishing here or how testing interneurons response to specific guidance cues vs ‘dissecting the role of specific signalling pathways’ can be addressed in a boyden assay
Reply:
We simplified the first sentence as following (line 187): “to study the effects of single guidance cues on interneuron migration or to dissect the role of specific signaling pathways”
Line 198: media changes
Line 202: specify here why in utero electroporation is preferred for projection neurons in comparison to interneurons (technical difficulty in targeting MGE)
Line 215: to label specific neuronal populations
Line 219: made up of different cyclical phases
Line 232: cells (hESCs) integrate into murine cortical slices
Line 286: distance travelled by interneurons when combined with live-imaging
Line 289: cortical explant
Line 300: exits the cell cycle
Line 307: electron microscopy
Line 319: insights into tangential migration in vivo to be gained
Reply:
All these small edits have been included.
Line 364-365: not clear why it is challenging to quantify interneurons which have reached the cortical plate – emphasise that they spread tangentially so requiring whole forebrain to be analysed?
Reply:
We thank the reviewer for this comment and we edited the corresponding section. See line 320-321: “since interneurons are spread throughout the whole forebrain”
Line 369: Through in-utero electroporation combined with live imaging,
Line 396: centrin to track centrosomes
Line 399: and translocate their nuclei upwards
Line 434: nucleus-centrosome uncoupling
Line 438: interactions
Line 444: Introduce abbreviation for double strand DNA break here (DSB)
Reply:
All the previous small comments have been taken into account and sentences edited.
Lines 458-459: If the word-limit allows, elaborate on the controls which need to be carried out for the Crispr/Cas9 technology .
Reply:
We thank the reviewer and we added a small sentence to explain better this concept (line 530): “ including Cas9-negative control and sequencing of amplicons containing potential off-target loci”.
Line 464: FRET technique is not introduced
Reply:
We added at line 538-544: “The principle of FRET, whereby energy transfer occurs between two fluorophores in close proximity, has been employed to design intramolecular probes, such as the one for RhoA. In this probe construct, RhoA, conjugated to one fluorophore (CFP), is also fused to the rhotekin binding domain (RBD), which in turn is conjugated to a different fluorophore (YFP). Activation of RhoA determines binding to RBD through an intramolecular conformational change that brings the two fluorophores close to each other, thus increasing FRET efficiency signal.”
Line 483-484: Unclear what is meant by ‘in-vitro assays do not allow tracking of cell migration in real-time’ as the stripe assays can be combined with live-imaging? Or do you mean that it’s not temporally physiologically-relevant?
Reply:
We agree that this sentence was misleading and changed it by specifying that “in vitro assays such as transwell assays ….” (line 562)
Line 501: Should mention that the advent of CRISPR/Cas9 technology is accelerating the time frame of generating transgenic mice
Reply:
We thank the reviewer and edited the sentence accordingly (lines 581-584): “Importantly, the advent of CRISPR/Cas9 technology is accelerating the time frame of generating transgenic mice and will therefore become an essential complement to in utero electroporation results.”
Other small changes throughout manuscript:
Use italics for terms in-vitro, in-vivo
Use adjective plasmid DNA not plasmidic DNA
Use adjective neuronal not neuron
Reply:
We made changes accordingly.
Reviewer 3 Report
The manuscript from Azzarelli and colleagues is a review focusing on the main techniques used to study neuronal migration in the developing cerebral cortex. A concise introduction presents the reader with general aspects of cortical neuronal migration. In vitro, ex vivo and in vivo approaches are then detailed in what constitutes the core of the paper. A conclusion summarizes the strengths and limitations of each technique.
The manuscript is overall well written, organised and illustrated. It mostly targets readers with little experience in the field of cortical migration, and provide them with a relatively extensive list of technical approaches of interest. The methodological aspects, potential applications and limitations are clearly stated. The choice of the authors to focus on the cerebral cortex is understandable. However, since most of the techniques presented have been (or could be) used to study other regions of the nervous system (or even other tissues), this review could raise interest in a broader audience.
Remarks:
- In some instances, the introduction is a bit simplistic. E.g. lines 16-17: “two distinct neuronal populations, glutamatergic and GABAergic neurons, are generated”. Although this is not a review on neuronal diversity, the authors should avoid implying that only two cell types undergo migration in the developing cortex. Not all glutamatergic neurons migrate radially, as exemplified by Cajal-Retzius cells, cortical plate transient neurons and subplate neurons, which are all tangentially migrating and glutamatergic (see Barber & Pierani 2015 for a review). Similarly, not all cortical interneurons derive from the ganglionic eminences as stated lines 90-91 (for example, Gelman et al. 2009 described interneurons migrating from the POA to the cortex). Please amend the text accordingly.
- I was a bit puzzled by the fact that mouse genetics seems not to be considered as a technique in itself by the authors. Most of our knowledge on cortical migration was actually gathered through the analysis of mouse mutants (loss-of-function, genetic tracing…). Perhaps a few words on the reasons of such an omission would be beneficial to the paper.
- Line 429-435: the case of Dcx is actually perfect to discuss the differences between acute and chronic loss-of-function. Contrary to shRNA-mediated knockdown (ref. #73 cited by the authors), Dcx knockout animals display no obvious radial migration phenotype in the neocortex (Kappeler et al., 2006). I encourage such an addition.
- A few lines could be added regarding the use of graft experiments that, combined with organotypic cultures, can prove very powerful to discriminate between cell-autonomous and non-autonomous mechanisms. Multiple examples can be found in the literature, notably from the lab of Oscar Marin.
Minor points:
- One of the limitation of in utero electroporation that is not mentioned in the manuscript is the difficulty to target the earliest born neurons. To circumvent such an issue, electroporation can been applied to embryos ex utero and followed by whole embryo culture (Osumi & Inoue 2001).
- Regarding the difficulty to target specific progenitors by in utero electroporation (lines 505-510), it is worth mentioning that the use of triple-electrodes (dal Maschio et al., 2012 ; Szczurkowska et al., 2016) may allow one to increase the precision of electroporation, especially to target regions such as the prefrontal cortex or visual cortex. In addition, the electroporation of plasmids containing loxP sites in Cre-expressing mouse lines can allow precise spatio-temporal control of the targeting.
- Lines 369-370: Pioneer work using time-lapse imaging on cortical preparations actually preceded the development of in utero electroporation (O’Rourke et al., 1992).
Line 351: the very first postmitotic cortical neurons are not found before E10
Author Response
We thank the reviewer for all his/her constructive comments. Please find detailed responses to each comment below:
The manuscript from Azzarelli and colleagues is a review focusing on the main techniques used to study neuronal migration in the developing cerebral cortex. A concise introduction presents the reader with general aspects of cortical neuronal migration. In vitro, ex vivo and in vivo approaches are then detailed in what constitutes the core of the paper. A conclusion summarizes the strengths and limitations of each technique.
The manuscript is overall well written, organised and illustrated. It mostly targets readers with little experience in the field of cortical migration, and provide them with a relatively extensive list of technical approaches of interest. The methodological aspects, potential applications and limitations are clearly stated. The choice of the authors to focus on the cerebral cortex is understandable. However, since most of the techniques presented have been (or could be) used to study other regions of the nervous system (or even other tissues), this review could raise interest in a broader audience.
Remarks:
In some instances, the introduction is a bit simplistic. E.g. lines 16-17: “two distinct neuronal populations, glutamatergic and GABAergic neurons, are generated”. Although this is not a review on neuronal diversity, the authors should avoid implying that only two cell types undergo migration in the developing cortex. Not all glutamatergic neurons migrate radially, as exemplified by Cajal-Retzius cells, cortical plate transient neurons and subplate neurons, which are all tangentially migrating and glutamatergic (see Barber & Pierani 2015 for a review). Similarly, not all cortical interneurons derive from the ganglionic eminences as stated lines 90-91 (for example, Gelman et al. 2009 described interneurons migrating from the POA to the cortex). Please amend the text accordingly.
Reply:
We thank the reviewer and we changed the paragraph accordingly.
See lines 62-66: “Although the distinction in radially migrating glutamatergic neurons and tangentially migrating interneurons is particularly helpful for discussion purposes, it is important to note that several classes of glutamatergic neurons, such as Cajal Retzius cells, a subpopulation of subplate neurons and transient cortical neurons also migrate tangentially to disperse along the medio-lateral axis.”
And lines 111-113. “In addition, recent work has shown that roughly 10% of cortical interneurons originate from the Preoptic area (POA) and migrate long distances to reach the cortex.”
I was a bit puzzled by the fact that mouse genetics seems not to be considered as a technique in itself by the authors. Most of our knowledge on cortical migration was actually gathered through the analysis of mouse mutants (loss-of-function, genetic tracing…). Perhaps a few words on the reasons of such an omission would be beneficial to the paper.
Reply:
We thank the reviewer for this comment and we added a sentence to explain why we did not discuss this in detail. See lines 133-138: “Conventional and conditional deletion of gene function, combined with genetic fate mapping in transgenic mouse models has strongly contributed to our understanding of the routes of migration and the molecules involved in the regulation of this process. However, since this review focuses on the technical details of the methodological approaches to investigate migration, we will not describe transgenic models here and we refer the reader to other reviews.”
Line 429-435: the case of Dcx is actually perfect to discuss the differences between acute and chronic loss-of-function. Contrary to shRNA-mediated knockdown (ref. #73 cited by the authors), Dcx knockout animals display no obvious radial migration phenotype in the neocortex (Kappeler et al., 2006). I encourage such an addition.
Reply:
We included this comment in lines 503-505: “It is important to note that Dcx knockout mice do not show overt migration abnormalities, thus highlighting the importance of acute knockdown via in utero electroporation to overcome compensation and study gene function.”
A few lines could be added regarding the use of graft experiments that, combined with organotypic cultures, can prove very powerful to discriminate between cell-autonomous and non-autonomous mechanisms. Multiple examples can be found in the literature, notably from the lab of Oscar Marin.
Reply:
We thank the reviewer for this comment and we edited accordingly this section. In llines 304-308, we added: “Rescue experiments also provide an efficient tool to distinguish cell-autonomous and non cell-autonomous mechanisms. An alternative method to assess cell-autonomous effects is based on graft experiments combined with organotypic slices: for instance, MGE-derived GFP-positive neurons are transplanted on WT or mutant organotypic slices in order to track perturbations in their migration.”
Minor points:
One of the limitation of in utero electroporation that is not mentioned in the manuscript is the difficulty to target the earliest born neurons. To circumvent such an issue, electroporation can been applied to embryos ex utero and followed by whole embryo culture (Osumi & Inoue 2001).
Regarding the difficulty to target specific progenitors by in utero electroporation (lines 505-510), it is worth mentioning that the use of triple-electrodes (dal Maschio et al., 2012 ; Szczurkowska et al., 2016) may allow one to increase the precision of electroporation, especially to target regions such as the prefrontal cortex or visual cortex. In addition, the electroporation of plasmids containing loxP sites in Cre-expressing mouse lines can allow precise spatio-temporal control of the targeting.
Reply:
We thank the reviewer and we discussed this point at section 3.
See lines 588-594: “It is for example still challenging to target distinct ventral subdomains, which are contiguous to each other, but give rise to different types of striatal and cortical interneurons. In order to increase the precision of the targeted site, the use of triple electrodes has recently reported impressive results. Other types of neurons that are difficult to target with electroporation are the earliest born neurons, mainly because of the size of the embryos at these early stages. Therefore, ex vivo electroporation followed by whole embryo culture is generally preferred.”
Lines 369-370: Pioneer work using time-lapse imaging on cortical preparations actually preceded the development of in utero electroporation (O’Rourke et al., 1992).
Reply:
We thank the reviewer and added this sentence, line 423-424: “Pioneer work from O’Rourke and colleagues using time-lapse imaging in cortical slices allowed visualization of the morphological changes that cortical neurons undergo during their migration.”
Line 351: the very first postmitotic cortical neurons are not found before E10
Reply:
We thank the reviewer and we amended it. Substitution of E9 with E10 (line 403).